# Posterior Polymorphous Corneal Dystrophy in a Patient with a Novel *ZEB1* Gene Mutation

**DOI:** 10.3390/ijms24010209

**Published:** 2022-12-22

**Authors:** Eva Fernández-Gutiérrez, Pedro Fernández-Pérez, Ana Boto-De-Los-Bueis, Laura García-Fernández, Patricia Rodríguez-Solana, Mario Solís, Elena Vallespín

**Affiliations:** 1Department of Ophthalmology, La Paz University Hospital, 28046 Madrid, Spain; 2Molecular Genetics Section, Medical and Molecular Genetics Institute (INGEMM) IdiPaz, La Paz University Hospital, 28046 Madrid, Spain; 3Biomedical Research Centre in the Rare Diseases Network (CIBERER), Carlos III Health Institute (ISCIII), 28029 Madrid, Spain; 4Molecular Ophthalmology Section, Medical and Molecular Genetics Institute (INGEMM) IdiPaz, La Paz University Hospital, 28046 Madrid, Spain; 5Clinical Bioinformatics Section, Medical and Molecular Genetics Institute (INGEMM) IdiPaz, La Paz University Hospital, 28046 Madrid, Spain

**Keywords:** posterior polymorphous corneal dystrophy, keratoconus, iridocorneal endothelial syndrome, *ZEB1*, confocal microscopy

## Abstract

Posterior polymorphous corneal dystrophy (PPCD), a rare, bilateral, autosomal-dominant, inherited corneal dystrophy, affects the Descemet membrane and corneal endothelium. We describe an unusual presentation of PPCD associated with a previously unknown genetic alteration in the *ZEB1* gene. The proband is a 64-year-old woman diagnosed with keratoconus referred for a corneal endothelium study who presented endothelial lesions in both eyes suggestive of PPCD, corectopia and iridocorneal endothelial synechiae in the right eye and intrastromal segments in the left eye. The endothelial count was 825 in the right eye and 1361 in the left eye, with typical PPCD lesions visible under specular and confocal microscopy. In the next generation sequencing genetic analysis, a heterozygous c.1A > C (p.Met1Leu) mutation was found in the *ZEB1* gene (*TCF8*). The PPCD3 subtype is associated with corneal ectasia, and both can appear due to a pathogenic mutation in the *ZEB1* gene (OMIM #189909). However, our patient had a previously unreported mutation in the *ZEB1* gene, which mediates the transition between cell lines and provides a pathogenic explanation for the epithelialisation of the corneal endothelium, a characteristic of PPCD.

## 1. Introduction

Posterior polymorphous corneal dystrophy (PPCD) affects the Descemet membrane and corneal endothelium, occurs more frequently bilaterally and asymmetrically and has mainly autosomal dominant inheritance [1,2,3]. There are unilateral isolated cases with a similar phenotype but without the inheritance pattern [1]. The phenotypic spectrum of PPCD is wide, even among members of the same family; patients might remain asymptomatic throughout their entire lives or might require keratoplasty at a young age [2].

The main endothelial corneal signs of PPCD are greyish lesions of varying shape and size, as well as vesicles and “rail tracks”. In rare cases, PPCD can be accompanied by iridocorneal peripheral adhesions, iris atrophy and corectopia. Over the years, glaucoma can appear due to abnormal endothelium growth over the angle [1,3,4]. Pathologically, there are several epithelial lineage-like features, such as stratified cell organisation, desmosomal intracellular junctions and proliferation of cells expressing the cytokeratins CK7 and CK19^5^, typical of epithelial lineage cells, which replace the hexagonal endothelial cells, producing abnormalities in the Descemet membrane [1,3,4,5,6].

The biomicroscopy findings are similar to those found in iridocorneal endothelial syndrome, which can also present endothelial abnormalities, synechiae and corectopia [1,7] (Appendix A: Appendix A).

The specular microscopy characteristic of PPCD is the endothelial rail tracks, also known as “snail tracks”, which are dark areas in the form of a band that encloses a number of smaller clear cells. The areas have irregular scalloped edges visible by slit-lamp examination. A certain degree of polymegathism and pleomorphism can also be found [1,3].

PPCD is a genetically heterogeneous disease with extremely variable expression in which three genes have been identified: *VSX1* (20p11.21), *COL8A2* (1p34.2-p32.3) and *ZEB1* (10p11.22) [3,4]. Mutations in these genes are characteristic of PPCD1, PPCD2 and PPCD3 subtypes, respectively. 

PPCD subtypes can only be reliably distinguished by genetic testing. Genetic diagnosis can provide important prognostic information that will aid clinical care. For example, patients carrying *OVOL2* (PPCD1) are more likely to require corneal graft surgery [2] and to develop secondary glaucoma compared to the other PPCD subtypes [8,9]. *ZEB1*-associated disease (PPCD3) is reported to be associated with significant corneal astigmatism, which must be managed in childhood to prevent amblyopia.

Individuals with *OVOL2* variants may become partially sighted and lifestyles must be adjusted accordingly. Individuals with *GRHL2* and *ZEB1* variants are less severely affected, but may not achieve the vision standard for driving [10].

*ZEB1* is a nuclear gene, which encodes a zinc finger transcription factor. It plays an important role in the process of differentiation and transcription process. The prevalence of *ZEB1* mutations (PPCD3) is approximately 33% in PPCD in certain geographical areas, such as the United Kingdom, Canada and the United States [5]. The clinical spectrum associated with the various mutation in patients with PPCD subtype 3 does not appear to be specific. The *ZEB1* p.Ser750X mutation and the frameshift mutation c.1578_1579insG (p.Val526fsX2) have been detected in patients with unilateral or bilateral corneal endothelial vesicles [11,12], while the phenotype of family members with the *ZEB1* p.1Met→Val mutation present bilateral disease, with the characteristic vesicular, band or diffuse appearance [13].

We present the case of a patient with a rare biomicroscopic manifestation of PPCD, associated with keratoconus, in whom the mutation NM_030751.6:c.1A > C (p.Met1Leu) was detected for the first time in heterozygosity in the *ZEB1* gene (OMIM #189909) *(TCF8*).

## 2. Case Presentation

### 2.1. Clinical Case

A 64-year-old woman was diagnosed with keratoconus in both eyes in October 2009 and was implanted intrastromal rings in the left eye in January 2013. She had no known systemic disease, and her only son (aged 36 years) presented no systemic or ocular disease.

### 2.2. Clinical Findings

In the ophthalmological examination, her uncorrected visual acuity was 0.5 (+2) in the right eye and 0.5 (+1) in the left eye, improving to 0.63 in the right eye with the pinhole test and showing no improvement in the left eye. Refractometry showed +0.50 −1.00 64° in the right eye and −0.25 −0.75 133° in the left eye. Under biomicroscopy, the right eye showed focal opacification in the inferior temporal and superior temporal corneal endothelium, anterior iridoendothelial synechiae from 4 to 6 o’clock, as well as superonasal corectopia. In the left eye, an intrastromal segment was observed at 70% depth. The corneal endothelium presented with rail tracks in the paracentral area, together with patchy endothelial opacification. The left iris showed no abnormalities (Figure 1a–d). The lens and fundoscopy results were normal for both eyes. Intraocular pressure were 12 mm Hg in the right eye and 11 mm Hg in the left eye.

Topography revealed a K1 (flattest meridian) of 46.6 diopters (D), a K2 (steepest meridian) of 47.3 D at 93.4°, a pachymetry apex of 557 microns and a posterior elevation of +28 microns in the right eye. The left eye had a K1 of 48.0 D, a K2 of 48.7 D at 105.4°, pachymetry of 562 microns and a posterior elevation of +38 microns, which were considered a right posterior keratoconus and a left keratoconus. 

The specular microscopy study showed a decreased cellular endothelial count, polymegathism, and polymorphism of the endothelial cells and a number of vesicles. The Heidelberg confocal microscopy study revealed keratocytes in the posterior stroma with spindle nuclei and polymorphism, polymegathism, and giant endothelial cells, with a number of nucleated cells that lay in both eyes. In the left eye, we observed a hyporeflective crater-shaped lesion and a curvilinear, hyperreflective band lesion (Figure 2a–d).

### 2.3. Molecular Genetics

Genetic analysis of six genes associated with PPCD, included in a panel of 298 genes related to ophthalmological diseases, was conducted using the massive sequencing technique, which found the initiator codon variant *ZEB1* (OMIM #189909): NM_030751.6:c.1A > C (p.Met1Leu) in heterozygosity, which was confirmed by Sanger sequencing (Figure 3). 

We proposed an ophthalmological and genetic study of the patient’s descendants. The results of the biomicroscopy, specular microscopy and genetic analysis of the proband’s 36-year-old son were normal.

### 2.4. Material and Methods

This ophthalmological and genetic approach was conducted by the Ophthalmogenetics Multidisciplinary Unit at La Paz University Hospital, Madrid, Spain, according to the tenets of the Declaration of Helsinki upon approval by the ethics committee.

#### 2.4.1. Ophthalmological Evaluation

A complete ophthalmological evaluation of the proband and her son was performed, which included best corrected visual acuity, refractometry, biomicroscopy, topography (Pentacam, Oculus, Oftas S.r.l, Policoro, Italy), specular microscopy (NIDEK CEM 530, Barcelona, Spain) and Heidelberg confocal microscopy (Heidelberg Engineering GmbH, Dossenheim, Germany). 

#### 2.4.2. Genetic Analysis

Genomic DNA was isolated from leukocytes in peripheral venous blood samples in the preanalytical area of our institute using the commercial Chemagic Magnetic Separation Module I (Chemagen, PerkinElmer, Waltham, MA, USA). DNA concentrations were measured by spectrofluorometer quantification using a NanoDrop ND-1000 Spectrophotometer (ThermoFisher Scientific, Waltham, MA, USA). Paired-end libraries were created using 1 µg of genomic DNA with KAPA HyperPrep Kit (Roche, NimbleGen, Inc., Pleasanton, CA, USA) and hybridization with a KAPA HyperCapture Reagent Kit (Roche, NimbleGen, Inc., USA).

The strategy for screening mutations was based on the use of next-generation sequencing, implementing a customized panel (OFTv2.1) including 374 genes related to ophthalmological disorders with a suspected genetic cause (Appendix A). The sequencing was performed on the Illumina HiSeq 4000 platform (Illumina, Inc., San Diego, CA, USA). The data produced were aligned and mapped to the human genome reference sequence (GRCh37/hg19). 

The OFTv2.1 panel was designed with NimbleDesign software (Roche NimbleGen, Inc., Pleasanton, CA, USA): HG19 NCBI Build 37.1/GRCh37, the target bases covered 99.44% and the size was 988,113 Kb. The mean horizontal coverage was 98.94%, and the mean sequencing depth per nucleotide was 231.

The first analysis was performed by the Institute of Medical and Molecular Genetics (INGEMM) Clinical Bioinformatics team, who developed an analytical algorithm that identifies point polymorphisms (SNP) and insertions and deletions of small DNA fragments inside the capture regions that are included in the next-generation sequencing panels. The system comprises a sample pre-processing step, alignment of reads to a reference genome, identification and functional annotation of variants, and variant filtering. All these steps employ open tools widely used in the scientific community, as well as proprietary tools. Furthermore, all phases are designed in a robust manner and include statistical parameters that provide information on the status of the process and the convenience of continuing with the analysis. This system allows for the monitoring of the process and the appropriate quality controls to issue a reliable report on the aforementioned variants. Lastly, the system backs up the raw and processed data, which are stored in a database using encrypted and anonymized records to preserve patient confidentiality.

The bioinformatics analysis was performed by the Clinical Bioinformatics Unit of the INGEMM centre using the following software tools: trimmomatic-0.36, bowtie2-align v2.0.6, picard-tools 1.141, samtools v1.3.1, bedtools v2.26 and GenomeAnalysisTK v3.3-0. The databases employed were dbNSFP v3.5, dbSNP v151, ClinVar date 20180930, ExAC-1, SIFT ensembl 66, Polyphen-2 v2.2.2, MutationAssessor release 3, FATHMM v2.3, CADD v1.4 and dbscSNV1.1. Below genotype-phenotype correlation was carried out. The pathogenic clinical significance of variants found in the patients was evaluated by employing the following databases: Varsome, Franklin Genoox, PubMed, the University of California Santa Cruz Genome Browser Home, GnomAD, LOVD and ClinVar. Figure 4 has been created with BioRender.com. 

## 3. Discussion

We present the clinical case of a woman with PPCD and keratoconus, with a previously undescribed mutation in the *ZEB1* gene. PPCD is inherited in an autosomal dominant form and is genetically heterogeneous, with mutations identified in three loci: the chromosome 20p (20p 11.21) *VSX1* gene mutation, characteristic of the PPCD1 subtype; the *COL8A2* gene (1p34.2-p32. 3) mutation, characteristic of the PPCD2 subtype; and the *ZEB1* gene mutation, characteristic of the PPCD3 subtype (2, 3). PPCD3 is often associated with corneal steepening [14,15].

The *ZEB1* gene (OMIM #189909) or zinc-finger homeodomain transcription factor 8 was altered in our patient. *ZEB1* is a nuclear gene which encodes a transcription factor involved in mediating epithelial and endothelial cell lineage transitions and is crucial during embryogenesis in the development of neural crest cell-derived structures such as the corneal endothelium [11,12]. The ZEB1 transcription factor is responsible for regulating the epithelial-mesenchymal transition, which involves the transformation of non-motile epithelial cells into cells with a mesenchymal phenotype that has the ability to migrate. ZEB1 induces the epithelial-mesenchymal transition by suppressing the expression of specific epithelial factors such as E-cadherin and is responsible for the reverse transition from mesenchyme to epithelium [5,8,9].

Taking into account the several epithelial-like features observed in the corneal endothelium of PPCD [9] and the molecular action of ZEB1, mutations in the *ZEB1* gene are likely to be responsible for the transposition of the endothelium into the epithelial phenotype characteristic of PPCD.

*ZEB1* mutations have been systemically linked with Alport syndrome and the progression of certain tumours. In ophthalmological cases, however, the gene has been associated with PPCD, Fuchs’ endothelial corneal dystrophy and keratoconus [8,13,15]. 

In 2013, Lechner et al. confirmed the genotype-phenotype correlation, such that mutations of the missense type in the ZEB1 protein resulted in Fuchs’ endothelial dystrophy and keratoconus, while the nonsense or LoF mutations of *ZEB1* produced a Stop codon, truncating the protein, causing PPCD3 [14].

In our patient, the isolated mutation was c.1A > C (p.Met1Leu) in heterozygosity in the *ZEB1* gene, a mutation not previously described in the literature or databases (Figure 4). There has been only one patient with a pathogenic mutation described in the same nucleotide, but the change is guanine instead of a cytosine: c.1A > G (p.Met1Val) [16]. Although the resulting amino acid is different, given that both variants change the translation initiator methionine codon, the resultant protein is described as p.Met1?, using a question mark to signify that it is not known if the loss of Met1 means that all protein translation is completely prevented or if an abnormal protein is produced using an alternate methionine.

**Figure 4 ijms-24-00209-f004:**
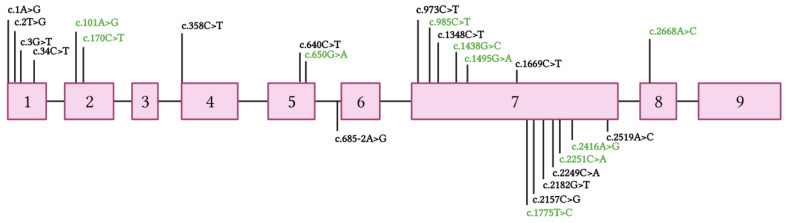
Point mutations classified as pathogenic and variant of uncertain significance (VUS) by the American College of Medical Genetics and Genomics (ACMG) [17] observed in *ZEB1* exons up to November 2022 [18]. Variants classified as pathogenic are shown in black. Variants classified as VUS are shown in green. Any point mutation classified as “likely pathogenic” has been reported in the Leiden Open Variation Database (LOVD) [18]. The reference transcript employed is NM_030751.6.

An LoF intolerant (pLI) score estimates the probability that a given gene is intolerant to haploinsufficiency [19]. A pLI ≥ 0.9 is widely used in research and in the clinical interpretation of cases with Mendelian inheritance, and genes with this score are considered to be extremely intolerant to LoF. The pLI score of the *ZEB1* transcript NM_030751.6 is 0.994, indicating high intolerance to LoF variants. Therefore, according to the ACMG guidelines (Table A2), our mutation is a null variant in a gene where LoF is a known mechanism of disease (PVS1). In addition, its frequency in the gnomAD population databases is extremely low (PM2). Three other variants in the Met1 position [13,20], classified as pathogenic by ACMG, have been reported in association with PPCD (PS1). Computational prediction tools for predicting the conservation of the variant (GERP++) [21] and algorithms developed to predict the effects of the variant on protein structure and function (Panther classification system) [22] suggest that this variant has been conserved over time (GERP++ score = 4.12) and is, therefore, likely to have a deleterious effect (PP3). Given the evidence, the variant is considered likely to be pathogenic and, therefore, associated with endothelial dystrophy in our patient. 

This genetic disorder is transmitted following an autosomal dominant inheritance pattern, meaning that the proband’s children have a 50% probability of inheriting the pattern. However, these genetic disorders present variable expression, and not all individuals with the disorder develop the same symptoms, the same severity, or the same clinical progression.

## 4. Conclusions

The PPCD3 subtype is associated with corneal ectasia, and both can appear due to a pathogenic mutation in the *ZEB1* gene. However, our patient had a previously unreported mutation in the *ZEB1* gene, which mediates the transition between cell lines and provides a pathogenic explanation for the epithelialisation of the corneal endothelium, a characteristic of PPCD.

## Figures and Tables

**Figure 1 ijms-24-00209-f001:**
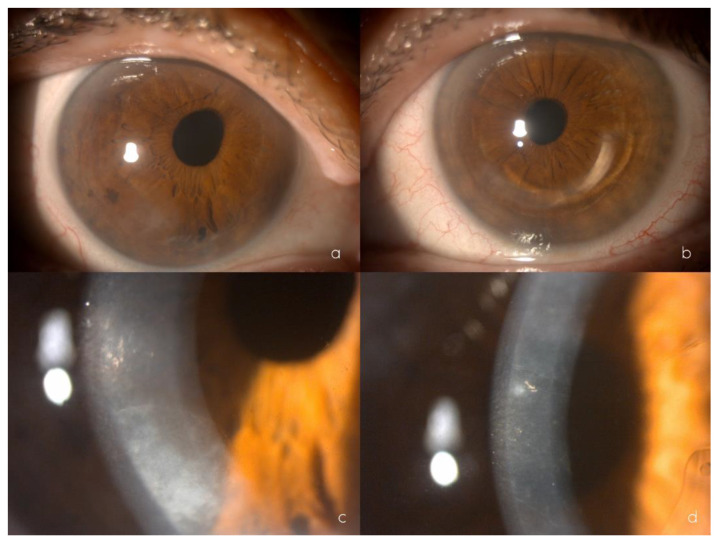
Biomicroscopy showing mild eccentricity of the pupil towards the upper nasal sector of the right eye (**a**), inferior intrastromal segment of the left eye (**b**), and inferior corneal endothelial opacification of the right eye (**c**), greyish polymorphic endothelial opacifications at different corneal locations and central endothelial rail tracks of the left eye (**d**).

**Figure 2 ijms-24-00209-f002:**
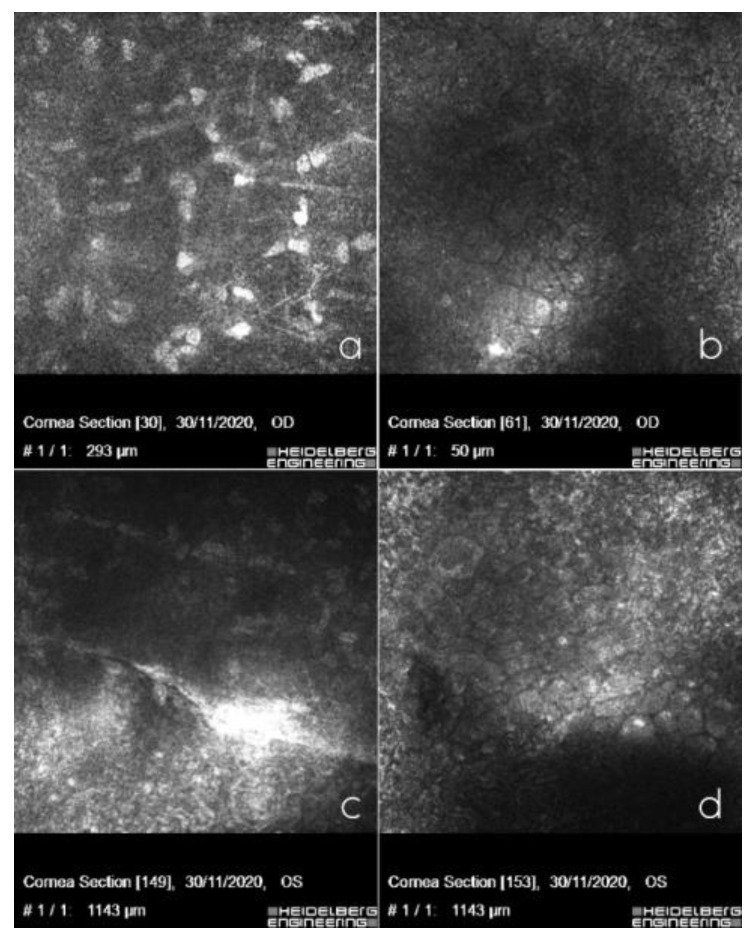
In vivo confocal microscopy showing posterior stromal keratocytes with elongated spindle-shaped nuclei in the right eye (**a**), polymorphism, polymegathism, giant endothelial cells with some nucleated cells in the right eye (**b**), curvilinear, hyperreflective banded lesion in the left eye surrounded by hyperreflective keratocytes (**c**) and hyporeflective vesicular lesions, in the form of a crater with hyperreflective deposits around the lesions in the left eye (**d**).

**Figure 3 ijms-24-00209-f003:**
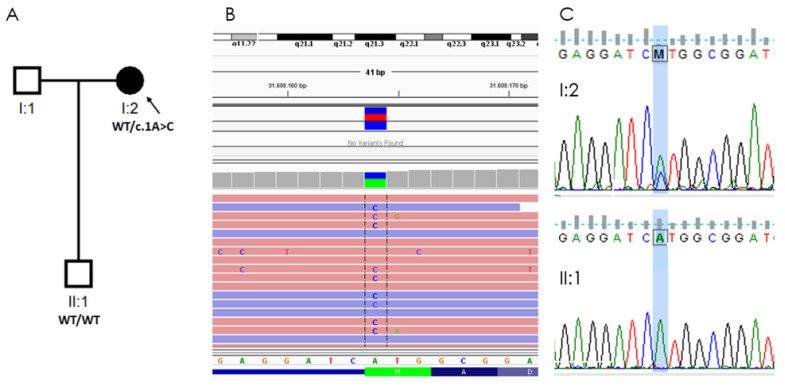
Genetic findings in a woman with PPCD and keratoconus and her healthy son. (**A**) Pedigree of the family, individual I:2 carries the loss-of-function (LoF) *ZEB1* mutation: NM_030751.6:c.1A > C (p.Met1Leu). (**B**) NM_030751.6:c.1A > C (p.Met1Leu) was detected in the proband by exome sequencing from leucocyte-derived DNA (visualized in Integrative Genomics Viewer, aligned to Genome Reference Consortium Human Build 37). (**C**) Mutation confirmed by Sanger sequencing in proband (I:2) and segregation study in her son (II:1).

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
