# Peer review of "Posterior Polymorphous Corneal Dystrophy in a Patient with a Novel ZEB1 Gene Mutation"

_ijms, 2022, doi:10.3390/ijms24010209_

Round 1

Reviewer 1 Report

They performed panel NGS for PPCD, and found novel ZEB1 gene mutations.

The content is novel, however, the presentation should be more skillful including genetic finding to support novel mutations.

1.English corrections and typos correction is required. For example, line 89 : “Display quotation – as needed.” Line 175 keratin 1911? Line 160: A 2013 study? Line 187 created with Bioreder.com described in method, not figure legend. Line 27, nucleotide sequencing is NGS? Or Sanger sequencing?

2. According to merican College of Medical Genetics and Genomics (ACMG) guideline 2022, classification should be several classificatory categories (i.e., “pathogenic”, “likely pathogenic”, “uncertain significance”, “likely benign”, and “benign” In Figure 4 only 2 categories VUS and pathogenic.

3. The reviewer regards the contents of PPCD subtype 3 should be described in detail in abstract and introduction and table1. The author found that patients with PPCD subtype3 presented novel mutations.

4. The introduction should be focus on PPCD and PPCD genes not difference diagnosis with ICE syndrome. In this case, bilaterality itself and keratoconus, the ICE syndrome is not related to patients. The table 1 is adopted from text book? In case, it provided as reference.

5. The authors should focus on the explanation of the mechanism in terms of loss of function of the newly founded mutation. More intensive revision required for better understanding for key finding presentation.

6. Gene id required OMIM id, transcript (reference mRNA sequence) requires NM numbers and functional category in text.

 7. A more detailed discussion of PPCD subtypes or causation, rather just ICE syndrome and differential diagnosis, would assist improve the paper's quality.

Reviewer 2 Report

In the manuscript, authors present a ZEB1 gene mutation in PPCD, these are some of my comments:

The phrase “…Display quotations of over 40 words, 89 or as needed... “is incomprehensible.

Please include Genomic Evolutionary Rate Profiling.

In silico analysis of the mutated protein should  be included.

A couple of phrases of gene characteristics would be useful.

The major concern is that the manuscript only presents a simple mutation in PPCD. The manuscript requires complementary data before being considered for publication.

Round 2

Reviewer 1 Report

Dear authors,

The reviewer regards that it looks a well-revised manuscript for the PPCD-related novel mutation.

Reviewer 2 Report

No comments